# Endoplasmic Reticulum Stress Provocation by Different Nanoparticles: An Innovative Approach to Manage the Cancer and Other Common Diseases

**DOI:** 10.3390/molecules25225336

**Published:** 2020-11-16

**Authors:** Amjad Ali Khan, Khaled S. Allemailem, Ahmad Almatroudi, Saleh A. Almatroodi, Ali Mahzari, Mohammed A. Alsahli, Arshad Husain Rahmani

**Affiliations:** 1Department of Basic Health Sciences, College of Applied Medical Sciences, Qassim University, Buraydah 52571, Saudi Arabia; k.allemailem@qu.edu.sa; 2Department of Medical Laboratories, College of Applied Medical Sciences, Qassim University, Buraydah 52571, Saudi Arabia; aamtrody@qu.edu.sa (A.A.); smtrody@qu.edu.sa (S.A.A.); shly@qu.edu.sa (M.A.A.); ah.rahmani@qu.edu.sa (A.H.R.); 3Department of Laboratory Medicine, Faculty of Applied Medical Sciences, Albaha University, Albaha 65527, Saudi Arabia; amoosa@bu.edu.sa

**Keywords:** endoplasmic reticulum stress, nanoparticles, ER stress mediated diseases, drug nanoformulation, anticancer drugs

## Abstract

A proper execution of basic cellular functions requires well-controlled homeostasis including correct protein folding. Endoplasmic reticulum (ER) implements such functions by protein reshaping and post-translational modifications. Different insults imposed on cells could lead to ER stress-mediated signaling pathways, collectively called the unfolded protein response (UPR). ER stress is also closely linked with oxidative stress, which is a common feature of diseases such as stroke, neurodegeneration, inflammation, metabolic diseases, and cancer. The level of ER stress is higher in cancer cells, indicating that such cells are already struggling to survive. Prolonged ER stress in cancer cells is like an Achilles’ heel, if aggravated by different agents including nanoparticles (NPs) may be exhausted off the pro-survival features and can be easily subjected to proapoptotic mode. Different types of NPs including silver, gold, silica, graphene, etc. have been used to augment the cytotoxicity by promoting ER stress-mediated cell death. The diverse physico-chemical properties of NPs play a great role in their biomedical applications. Some special NPs have been effectively used to address different types of cancers as these particles can be used as both toxicological or therapeutic agents. Several types of NPs, and anticancer drug nano-formulations have been engineered to target tumor cells to enhance their ER stress to promote their death. Therefore, mitigating ER stress in cancer cells in favor of cell death by ER-specific NPs is extremely important in future therapeutics and understanding the underlying mechanism of how cancer cells can respond to NP induced ER stress is a good choice for the development of novel therapeutics. Thus, in depth focus on NP-mediated ER stress will be helpful to boost up developing novel pro-drug candidates for triggering pro-death pathways in different cancers.

## 1. Introduction

The endoplasmic reticulum (ER) comprises a vast membranous network within a eukaryotic cell, which performs different functions such as protein folding and processing, lipid biosynthesis, and calcium storage, etc. This organelle acts as the major assembly point for almost all the secretory and integral membrane proteins. Within the ER, the nascent polypeptides enter through translocation and are properly folded by different covalent and non-covalent modifications and are then assembled to achieve their higher order complexities [1,2]. The presence of different chaperons within the ER lumen like heat shock protein homologues (hsp-40, -70, -90), ER lectins (calreticulin, calnexin) and thiol oxidoreductases such as protein disulfide isomerase (PDI), serpins, binding immunoglobulin protein (BiP), etc. makes this organelle a unique and ideal system for proper protein folding [3]. Different types of co-translational and post-translational modifications occur inside the polypeptide chains within the ER lumen and are shaped as native 3-dimensional proteins. These final form proteins are transported across the ER for their further final destinations. In addition, this organelle also performs its role in checking any improperly folded proteins, which are selectively identified and marked for destruction. In this way, only the properly folded proteins, which pass the checkpoint of ER quality control, are allowed to exit. In addition to protein modifications, the ER is responsible of other multiple functions as vesicular traffic, biosynthesis of cholesterol, phospholipids, and calcium buffering [4].

The quality control system of ER, responsible for the proper shaping of proteins, is prone to influence by different intracellular as well as extracellular stimuli. Different factors affect this proper protein folding capacity, which includes oxidative stress, hypoxia, viral infection, aberrant Ca^2+^ regulation, glucose deprivation, environmental toxins, mutant protein expression, hypoglycemia, and aging, etc. [5]. In addition, ER stress is induced by several other factors like high sugar and high fat diet, and some drugs as bortezomib, viracept, celebrex, celecoxib, etc. Furthermore, several natural compounds (e.g., tunicamycin, thapsigargin, and geldanamycin) also induce this stress. To buffer the ER stress, a cell can use four different strategies like translational attenuation, UPR upregulation, ER compartmental proliferation, or finally programmed cell execution [6,7].

The redox-state of ER is closely linked with its protein-folding homeostasis. Any alteration of this redox-balance has a very high impact on disulfide bond formation within the ER lumen, where both the oxidizing and reducing reagents disrupt the protein folding, creating the ER stress [8]. At the time of oxidative protein folding, the thiol groups present on cysteines are oxidized and lead to the formation of disulfide bonds with the generation of hydrogen peroxide (H_2_O_2_). During the ER stress, this disulfide bond formation dysregulation results in greater reactive oxygen species (ROS) accumulation, resulting in oxidative stress (OS) (Figure 1). Furthermore, some UPR components like the C/EBP homologous protein (CHOP) can also contribute to this OS. The ER stress also results in mitochondrial dysfunctions, causing increased mitochondrial ROS formation. It has also been reported in many in vivo and in vitro models that the ER stress and OS emphasize each other positively in a feed-forward loop, leading to the interference of cellular functions and activating pro-apoptotic signaling [9].

Any type of disturbance, which hinders the normal functioning of the ER, results in upregulation of unfolded protein response (UPR), which initially aims to compensate for the damage. However, in some severe situations that shatter the ER function severely or for a prolonged time, the UPR signal triggers the cell death. This mechanism of ER stress to cell death remains enigmatic, as multiple participants have been described with clear transparency. Several common disease like diabetes, cardiac problems, ischemia, neurodegeneration, and cancer, etc. have been linked to ER stress mediated cell death [10].

In this review, we mainly aimed to focus on an overview of ER stress, its intervened diseases, the role of the physico-chemical properties of different NPs on biological systems as toxicological and therapeutic agents, and their role to induce ER stress. Almost no review article has been published thus far, which includes different metal and non-metal based NPs and their specific roles as ER stress inducing agents. This review may definitely provide a significant information about diverse NP mediated ER stress as an innovative approach for the management of disease therapy including cancer.

## 2. An Overview of Endoplasmic Reticulum (ER) Stress

Different factors that affect the normal activity of ER like protein disulfide bond formation and glycosylation are affected by overexpressed mutated proteins, which results in its stress. To respond to ER stress properly, the eukaryotic cells have adapted a group of signal transduction pathways, which are collectively termed as unfolded protein response (UPR). UPR is a homeostatic signal transduction network that coordinates the retrieval of ER functions. A cell undergoes apoptosis, if there is any failure that results in the adaptation of ER stress. Insight knowledge has led to the recognition of regulatory mechanisms and crosstalk signaling, which involves three branches of UPR. These three branches are initiated by inositol-requiring protein 1α (IRE1α), protein kinase RNA-like ER kinase (PERK), and activating transcription factor 6 (ATF6) (Figure 1). The relationship between UPR and the cellular decision about life or death during the ER stress is fascinating and has led to special interests to find the link between UPR signaling and different human diseases including cancer. Here in this section of the review, we focus in brief on the molecular mechanism of UPR signaling that leads to cell death during ER stress [11].

The most phylogenetically preserved UPR signaling occurs through IRE1α. The IRE1α gets oligomerized in response to the ER stress and activates the kinase and endoribonuclease functions, which are present on its cytosolic domain [12,13]. The cytoprotective output of IRE1α is mediated by the specific splicing of Xbp1 mRNA. The spliced Xbp1 encrypts a strong transcription factor, whose gene targets code the proteins, which enhance the protein folding capacity of ER and the degradation of misfolded proteins. This signal transduction pathway protects the cell by reducing the stimulus that ultimately leads to ER stress [14,15].

As an additional function to ER stress, the IREIα can promote cell death through the activation of c-Jun N-terminal kinase (JNK). The IRE1α kinase domain binds with the adapter molecule TRAF2, which leads to the activation of apoptosis signal regulating kinase (ASK1) that phosphorylates and activates JNK. The activation of JNK triggers the cell death in response to UV irradiation or TNFα receptor activation [16,17].

The cell death is also promoted by IRE1α signaling through the activation of caspases, which act as the actual apoptosis effectors. The tumor necrosis factor-receptor associated factor 2 (TRAF2), an adapter molecule that interacts with procaspase-12, and this interface gets disrupted by the ER stress, which promotes the conversion of procaspase-12 to the active enzymes [18] (Figure 1).

The signal transduction of UPR mediated ER stress, through the PERK, also provokes the proapoptotic effects after its activation. The PERK protein also bears an ER-lumenal peptide domain that is homologous as IRE1α. This domain monitors the proper protein folding within the ER lumen. The ER stress also leads to the PERK oligomerization like that of IRE1α, and this leads to the activation of kinase activity present in its cytosolic domain [19]. Eukaryotic initiation factor 2α (eIF2α) is the target for this kinase activity, which is a ubiquitous cofactor. This cofactor is needed for the assembly of 80S ribosomal subunits to inaugurate the protein synthesis [20]. The PERK mediated phosphorylation of eIF2α inhibits its functions and leads to the decline of protein synthesis as the ribosomal subunits do not assemble efficiently on mRNAs [21,22]. In this way, PERK signaling shields a cell from ER stress-mediated protein misfolding. This signaling of translational attenuation mediates a good protective advantage to the cells under stress. This signaling may also lead to cell death, if the protein synthesis drops below the level necessary to sustain vital activities.

To control the translational over-attenuation, the PERK signaling also leads to the activation of a regulatory protein phosphatase subunit GADD34. This promotes the dephosphorylation of eIF2α, which helps to restore the mRNA-ribosomal assembly [23]. However, these signal transduction mechanisms are still unclear as to whether the translational attenuation damages or protects a cell.

The third brand of UPR signal transduction during an ER stress is initiated by ATF-6α, which is a transmembrane stress sensor protein [24,25]. In contrast to PERK and IRE1α, which are ubiquitous, this type of ER stress transducer is expressed in a cell and tissue specific manner [26]. This ER transmembrane protein also possesses a stress-sensing ER luminal domain that is coupled through the transmembrane segment to the cytosolic transcription factor domain [27]. With the initiation of ER stress, the ATF-6α operates from the ER to the Golgi complex. In the Golgi complex, some specific proteases break the ATF-6α transmembrane domain and release the cytosolic domain [28]. This detached protein fragment translocates to the nucleus, where it acts as a transcription factor. This phenomenon upregulates the UPR target genes, which are overlapped with the genes activated by ATF-4 and XBP-1. This way, ATF-6α is believed to protect a cell from stress [29]. However, some findings also suggest that ATF-6α possesses some proapoptotic functions [30].

As the ER stress regulates the UPR, it can lead to cell death. It is not astonishing that such circumstances can also promote the protein misfolding or declined cellular abilities. To handle such misfolded proteins, the stress ultimately results in different cellular dysfunctions and diseases including cancer. Therefore, any inappropriate UPR activation may be harmful to a cell, leading to cell death. Next, we discuss in detail how this artificially induced ER-stress strategy can be a better therapeutic opportunity to cure an organism from different diseases including cancer.

## 3. Endoplasmic Reticulum Stress Mediated Diseases

Any type of perturbation within a cell that affects the normal functioning of the ER activates special signaling cascades that organize the adaptive and apoptotic responses. It is now well evidenced that prolonged ER stress leads to the development and progression of different diseases which include liver diseases, atherosclerosis, neurodegeneration, type 2 diabetes, and cancer, etc. A proper understanding of the molecular mechanisms of this ER stress response can be a potential strategy to treat such diverse diseases [31]. The ER stress mediated UPR contributes to different types of diseases including cancer, as listed in Table 1.

## 4. Endoplasmic Reticulum Stress and Cancer

As we know, cancer cells possess a higher growth rate and proliferation, so these cells demand an increased rate of protein folding and assembly within the ER. In addition to this, some tumor cells express additional mutant proteins that are not properly folded, which further leads to ER stress mediated UPR cascades. The initiation of malignancy is followed by poor vascularization, which leads to nutrient starvation around the tumor mass. The increased hypoxia and changes in the redox environment strongly induce the UPR signal transduction cascades. Recent evidence strongly suggests that cancer cells favor the UPR environment and this setting acts as an important survival pathways for such cells. Different types of cancers have been reported with a higher expression of ER chaperons and enhanced appearance of UPR markers. It has also been found that GRP78 expression is increased in hepatocarcinoma, adenocarcinoma, colon, and breast cancer cell lines [53]. The increased expression of GRP78 is believed to favor the cancer cell survival signals and also convenes the drug resistance. The pathologic grade and recurrence of cancer is also correlated with higher expression of GRP78 in patients suffering from liver, breast, prostate, gastric, and colon cancer [54].

In parallel to the GRP78 expression, the overexpression of XBP1 has been reported in different human cancers like hepatocellular carcinoma and breast cancer. In addition, overexpressing XBP1 in transgenic animal models has been reported to achieve plasma cell neoplastic transformation as well as the development of myeloma [55]. Even though many studies support the overexpression of UPR in human cancers and malignant animal models, a recent report showed the downregulation of UPR in prostate cancer mouse models [40]. Therefore, all these observations conclude that ER stress during cancer may be more complicated, as initially predicted.

It is now well documented that the usual drug resistance approach, experienced by the different types of cancer cells, can be overcome by targeting the ER stress signaling pathways. These anticancer drugs are believed to reduce the adaptation of tumor cells for inflammation, hypoxia, and angiogenesis [56]. In this regard, several anti-tumor drugs have recently been designed and studied, which directly act through ER stress pathways and affect cancer progression. However, the proper drug targets (cancer cells only) remain a challenge to the use of such powerful drugs [57,58].

## 5. Endoplasmic Reticulum Stress as a Novel Target to Fight against Cancer Cells

The controlled ER stress experienced by cancer cells employs their antiapoptotic functions, thus supporting cell survival and also enhancing the chemoresistance. Any type of aggravation exceeding the protective capacity of these cancer cells can switch on their proapoptotic module [59]. Therefore, targeting and enhancing the ER stress in cancer cells is emerging as a novel target for different anticancer drugs and metal nanoparticles. To understand the association between aggravated ER stress and apoptosis can be an innovative approach to combat different types of cancer.

The UPR is considered as a novel therapeutic target in different cancer cells, and several pharmacological agents induce impairment to UPR and lead to cell death [60,61]. The different pharmacological agents specified are 17-AAG, Bortezomib, and Brefeldin-A, etc., which have been recently used to induce UPR inhibition [62,63]. In-clinic chemotherapeutic compounds like cisplatin and doxorubicin have also been used to explore the new targets for inducing the ER stress.

In the recent past, different types of cytotoxic compounds have been engineered to target the ER, which often exhibit selectivity for some cancer cells compared to normal cells. These drugs have been successfully used to induce ER stress beyond the cancer cell capacity to acquire immunogenic cell death. Some of the potential ER stress inducing agents include metal complex NPs possessing redox activity, which appear as promising candidates to fight cancer [11].

Dozens of metal and non-metal based NPs have been engineered in the last two decades that are reported to kill different types of cancer cells through the induction of ER stress. Some of these NPs exhibit anticancer activity even at nanomolar concentrations, used either as in vitro or as in vivo conditions.

## 6. Physico-Chemical Characteristics of Nanoparticles (NPs) and Their Role on Biological Systems

A vast comprehension about the NPs physico-chemical characteristics and its interactions with biological systems is of significant importance. The different physical characteristics of NPs include their composition, shape, size, and surface chemistry, which play a great role in their biomedical applications. These NP fundamental properties play a great role in determining the biological kinetics, biomolecular signaling, transportation, and toxicity in both in vivo and in vitro studies [64].

First of all, a typical NP is synthesized chemically to manage its size and surface chemistry, is loaded with specific drugs and surface coated with some polymers, and is eventually administered into a cell culture or any animal model. Recent studies have indicated that the interaction of NPs with serum proteins and cell membrane receptors is specified by the NP basic design, which determines its cell uptake, organelle interaction, gene expression, and cyto-toxicity. The NPs interact with cell membrane in multiple ways depending upon the cell membrane ultra-structure and the NPs’ physico-chemical state [65] (Figure 1).

A NP is also designed to dictate the interaction between the ligands with the receptor targets. A NP can be synthesized by having multiple ligands to offer multivalent effects when it interacts with multiple receptors present on the cell surface. This results in more binding strength (avidity) compared to the sum of individual affinities. The density of ligands and its specific curvature, present on the surface of NPs, also contributes in overall avidity strength. The affinity of a ligand binding increases proportionally with the NP size. This phenomenon has been checked by studying the avidity between Herceptin to ErbB2 receptor as 10^−10^ M in solution, 5.5 × 10^−12^ M on a 10 nm diameter NP, and 1.5 × 10^−13^ M on a 70 nm NP [66]. However, other factors in addition to binding affinity may also participate in determining the biological effects. This phenomenon occurs when 40–50 nm AuNP induces its strongest downstream signaling via the ErbB2 receptor. In addition, NP design may also induce differential cell signaling compared with the free ligand in solution. This phenomenon is supported when Herceptin coated, 40–50 nm, AuNPs altered the cellular apoptosis by affecting the caspase enzyme activities [66].

Similarly, NPs conjugated with receptor-specific peptides may also improve their capability of angiogenesis induction [67]. The NPs can also lead to some unexpected phenomena in cell signaling. For example, intercellular adhesion molecule I (ICAM-I) coated NPs get internalized, which is an unusual finding because ICAM-1 is not known for triggering endocytosis [68]. A study has shown that carbon-NPs (14nm) interact with β1-integrins and epidermal growth factor receptors (EGFRs) and induce the activation of Akt signaling, thus leading to cell proliferation [69]. Furthermore, an additional complexity occurs when the NP-ligand complexes, which can also lead to the denaturation of proteins present on the cell surface. This denaturation can lead to altered NP receptor binding, increased nonspecific interactions, or provoked inflammation. This phenomenon is observed when lysozymes are attached to AuNPs, denature, and interact with other lysozyme proteins, producing protein-NP aggregation and finally inflammation [70,71].

## 7. Nanoparticles Used as Toxicological and Therapeutic Agents Including ER Stress

The NPs synthesized from different metals and non-metals have proven a great opportunity for cancer theranostics. Engineered NPs including silver, gold, copper, and copper oxide NPs, etc. have been reported to induce cytotoxicity, which triggers detectable toxicological changes through the generation of ROS [72,73]. However, the exact correlations between ROS production and ER stress response had not been clearly outlined in toxic assessment from different nanomaterials. Different researchers have indicated that some NPs can induce apoptosis through the activation of mitochondria-mediated pathways [74,75]. However, AgNPs have been found to induce apoptosis though the modulation of ER stress reactions. In addition to this, some recent findings have reported that some NPs can lead to the induction of ER stress by activating different cellular reactions, which include the initiation of the apoptotic and inflammatory pathways [72]. The AuNPs have some potential medical usage and have been worked out as efficient cellular ER stress elicitors [76]. Zinc oxide (ZnO) is an important engineered nanomaterial that shows some toxicity to some mammalian cells. The ZnONP dissolution within the cells leads to the release of toxic Zn^2+^ ions, which are capable of ROS generation [77]. The Ceria nanoparticles (CeNPs) exhibit some antioxidant activity as they reversibly bind oxygen and can switch between Ce^4+^ (oxidized) and Ce^3+^ (reduced) forms at the surface of CeNPs [78].

Some of the metal based NPs and engineered liposomes have therapeutic efficacy against many tumors due to their unique ability to kill only specific cancer cells [79,80]. Despite a great enthusiasm to use NPs for different biomedical applications, their advancement in clinical studies is comparatively slow because the adverse outcomes of their use during in vivo studies is not fully understood [81]. Therefore, there is an urgent requirement to properly investigate the NP induced toxicity mechanisms within the cells, as this strategy is very eagerly required to combat cancer. In addition, the proper mechanism of different NP action is required to ensure their safe use as well as the new design of more biocompatible NPs. In addition to the multiple use of NPs for commercial purposes, some other metalloid and non-metal NPs like silicon based NPs and carbon nanotubes (CNTs) have proven to have great potential as targeted-delivery drug nanocarriers [82].

Several specific mechanisms have been proposed to explain the toxicological role of NPs. Most of the mechanisms support oxidative stress as an important part of their toxicity. This is because various chemically active NPs induce the production of ROS, leading to oxidative damage [83]. Furthermore, some NPs directly provoke an inflammatory response and disturb immune system [84]. NPs have also been reported to dysfunction lysosomes and induce autophagy [85]. In recent years, a deeper look at the mechanism of action of some NPs has supported their role as inducing ER stress as a promising mechanism of NP induced cellular toxicity [86].

Recently, the nanoscale materials including different types of NPs have been effectively used to address different types of cancers in depth. The various types of NPs including lipidic and polymeric NPs and small molecule based supramolecular self-assemblies have been synthesized to precisely navigate to the ER, promote stress, and check the impairment of the UPR [87]. Graphene oxide (GO) based NPs have also emerged as novel candidates with an outstanding panoply of features [88]. These NPs are effectively biodegradable and biocompatible with unique surface modalities allowing for the stacking of drugs and conjugation of targeting moieties [89].

The synthesis of NPs faces a great challenge with regard to their stability and strength in different media. This challenge multiplies even more when NPs are engineered for use as payloads for different drugs, route of delivery, and as organelle targeting. However, there is an urgent need for effective nanoscale tools for the effective impairment of the adaptive UPR and induction of ER stress mediated apoptosis in cancer cells.

Here, we discuss the toxicological and therapeutic effects of some well-known NPs (Table 2) that have been used to induce ER stress as well in different in vivo and in vitro conditions.

### 7.1. Silver Nanoparticles

Different types of nanomaterials including silver nanoparticles (AgNPs) have been widely used to induce intracellular oxidative stress or ROS-mediated cytotoxicity. The treatment of cells with AgNPs also leads to membrane leakage, poor mitochondrial functions, and declined viability in different cell types like rat hepatocytes, germline stem, and neuroendocrine cells [144,145].

As AgNP induced cytotoxicity is related to oxidative stress, it is strongly evidenced that ER may perform an important role in AgNP mediated apoptosis. The key participants of ER stress include PERK, IRE1, and ATF6 [29]. ATF6 and the spliced form of XBP1 play a positive role in regulating the expression of different ER stress genes that include ER resident chaperons like GRP78/Bip and GRP94 [29,146]. The proapoptotic transcription factor CHOP/GADD153, which acts as a transcription suppressor of Bcl-2, can be induced by the joint ATF6 and PERK/ATF4 pathways [147,148]. The overexpression of CHOP induces cell death, while CHOP gene deletion results in cell death attenuation induced by ER stress [149]. Flow cytometry side scatter (FCM-SS) analysis has been previously used to determine the incorporation of AgNP within the cell cytosol. In addition, it has been found that AgNPs induce Ca^2+^ overloading within mitochondria, which indicates that Ca^2+^ homeostasis plays an important role in AgNP-induced apoptosis (Figure 1).

The dosage of different NP formulations is important in evaluating various adverse effects including ER stress. In mice, the cytotoxicity and genotoxicity have been evaluated at a single dosage of 25 mg/kg/day for three successive days with 15–100 nm AgNPs [150]. A study showed that AgNPs induce apoptosis via ER stress in cell models [151] as well as in the spleen and liver. Stress marker proteins like BiP and HSP70 are induced by a dose dependent process. The AgNP mediated organ damage is accompanied by ER stress sensor protein activation including PERK, IRE1, and CHOP proteins. It has been further reported that ER stress mediates a significant decrease in the DNA content of mice liver after AgNP feeding, suggesting the possible induction of apoptosis [152].

The AgNP mediated apoptosis mostly occurs in the liver, lung, kidney, and spleen, but also, no apoptosis has been seen in the heart and brain. Furthermore, a study has also reported that AgNP induced oxidative stress (OS) is demonstrated by the upregulation mRNA of HO1, SOD1, and GPX. In addition, the level of different inflammatory markers like IL-6 and TNF-α, the expression increases at higher concentration of AgNPs [152,153].

There is a little information available regarding the sensitivity of different tissues for AgNPs [154,155]. It has been observed that 16HBE possesses the highest sensitivity to AgNPs compared to human umbilical vein endothelial cells (HUVECs) and HepG2 cells. In parallel, the ER stress was responded significantly by 16HBE cells compared to the other two cell lines. The upregulation of different proteins in 16HBE cells was in parallel to the overexpression of different proteins in HUVECs by AgNP induced ER stress [72]. It also leads to the overactivation of different genes including HSPs like HSPA1b, HSPH1, and ER proteins Mdg1/ERdj4 (DNAJB9), a chaperone belonging to the HSP40 family [156]. The exposure of AgNPs also leads to the induction of ER stress marker genes like DNA damage-inducible protein 34 (GADD34/PPP1R15a). Furthermore, these NPs also lead to the overexpression of homocysteine-inducible, ER stress-inducible, ubiquitin-like domain member 1 (HERPUD1). Moreover, a significant expression of DDIT3/CHOP genes occur by AgNP exposure. The physicochemical properties of AgNPs and their specific role for inducing ER stress is further summarized in Table 3.

### 7.2. Gold Nanoparticles

Gold nanoparticles (AuNPs) are emerging as novel agents for cancer treatment and are being explored as drug carriers, radiosensitizers, and photothermal agents. The biological effects of X-irradiation efficiency is enhanced by high atomic number elements. In this regard, AuNPs have been used as a favorable radiosensitizer, in addition to iodine and iododeoxyuridine [157,158]. It has been found that the radiosensitization is very effective for both free cell suspension in the presence of 1% AuNPs (1.5–3.0 μm diameter) as well as for tumor cells injected with AuNPs [159]. The size of such AuNPs limits their use for intracellular medium, so the size reduction is the only choice for their specific organelle-targeting. Smaller AuNPs (1.9 nm diameter) have been found to be internalized and uniformly distributed within the transplanted tumor cell cytoplasm [160]. Moreover, it has also been observed that the accumulation of 13 nm AuNPs in B16F10 melanoma cells leads to apoptosis through enhanced radiosensitization [161].

The stability of gold nanosol has been dramatically enhanced by coupling it with polyethylene glycol (PEG). PEGylated AuNPs have been used to sensitize CT26 colorectal adenocarcinoma and EMT-6 breast cancer cells to different forms of ionizing radiations [162]. The PEGylated AuNPs have also been found to possess higher bioefficiency, however, the detachment of PEG leads to decreased dispersion stability. A unique PEGylated gold nanogel has been prepared that possesses a large payload capacity of AuNPs (8 nm diameter) [163]. This form of nanogel is made up of a cross-linked poly(2-[*N*,*N*-diethylamino]ethyl methacrylate) (PEAMA) gel core tethered with PEG chains. The tunneling electron microscopy (TEM) results revealed that the diameter of Au-nanogel particles was almost 106 nm, and approximately 15 AuNPs were calculated to be included in each PEGylated nanogel [163]. A general representation of PEGylated AuNPs is shown here in Figure 2.

This Au-nanogel has been found to possess exceptionally high dispersion stability with unique properties like reversible volume phase transition in response to ionic strength, temperature, and pH change [164].

Recently, it has also been observed that AuNPs show different biological effects like cell cycle regulation in addition to the production of ROS and the induction of apoptosis [165]. It has been reported that PEGylated phospholipid (PL) nanomicelles, even though not containing AuNPs, accumulated in ER and activated EPR, thus inducing ER stress, which ultimately leads to apoptosis [166]. The physicochemical properties of AuNPs and their specific role for inducing ER stress are further summarized in Table 3.

### 7.3. Iron Oxide Nanoparticles

Iron oxide nanoparticles (FeONPs) have been widely investigated for their novel biomedical applications. These applications include magnetic resonance imaging (MRI), cell tracking, magnetic transfections, tissue repair, detoxification of biological fluids, drug delivery, and hyperthermia treatment [167,168]. FeONPs have also been used for their efficient antifungal, antibacterial, and anticancer properties. These NPs have some promising properties such as low cost, ease of synthesis, biocompatibility, and are supramagnetic.

The different NPs from iron include maghemite (γ-Fe_2_O_3_), hematite (α-Fe_2_O_3_), and magnetite (Fe_3_O_4_)NPs. For the imaging of different diseases, ultra-small super paramagnetic FeONPs have been applied as an intravenous injection, which can be accumulated in spleen and liver. These NPs are used for the imaging of liver diseases such as cirrhosis, hepatitis, and hepatocellular carcinoma [169,170].

The genomic expression results have shown that ultrasmall paramagnetic FeONPs (USP-FeONPs) affects many signaling pathways of inflammatory response such as IL-6 release from the diseased cells [171]. These NPs lead to ER expansion, upregulate calcium ions, and induce ER stress in hepatocytes. It has also been observed that USP-FeONPs induce higher inflammation and cytotoxicity and also trigger IL-6 related inflammation, which is regulated by UPR and ATF4/PERK pathways. This indicates that ATF4/PERK pathways could be a specific and potential target to attenuate USP-FeONP-induced hepatic inflammatory response [172].

The cytotoxic effects of hematite NPs on Hekk293 cells have been observed by using 180 nm sized NPs at a dose of 100–500 μg/mL for one day exposure time where it was observed that these NPs did not show any toxicity at a lower dose (<300 μg/mL), but significant cytotoxicity was observed at a >350 μg/mL dosage. These hematite NPs induce the oxidative stress in Hek293 cells and lower the antioxidant capacity and the activity of antioxidant enzymes [173].

The hematite NPs (15–30 nm diameter) have been used to study their anticancer activity against HepG2 liver cancer cells [174]. The different concentrations from 50 to 1000 ng/mL were used to study this cytotoxicity in HekG2 cancer cells. The dimercaptosuccinic acid (DMSA)-coated supramagnetic FeONPs were used for their internalization and clearance studies within the HepG2 cells [175]. It was found that micropinocytosis uptake and clathrin-mediated internalization depends upon the particle size (Figure 1). It was reported that these NPs are accumulated in MDF-7 cells without any significant effects on cell morphology, ROS generation, and cell viability. It has been concluded from these studies that DMSA-coated supramagnetic FeONPs possess excellent biocompatibility to target breast cancer cells. Furthermore, their anti-tumor effect on MCF-7 cells suggests that these NPs may be used to target drug accumulation in cancer cells. However, the lower concentrations showed more efficiency in tumor cell disturbance, and lowering the therapeutic dose can reduce the side effects of chemotherapy [176].

The role of FeONPs in tumor cells has been further studied and it has been observed that tumor cells are effectively killed by heat production with the help of magnetic hyperthermia in whole tumor regions of the breast. The heating potential is functionalized by efficient NP cell internalization and the effects of the chemotherapeutic agent [177]. In addition, it has been reported that the therapeutic effects of magnetic hyperthermia in breast cancer could be strongly enhanced by the combination of MF66 functionalized with N6L and doxorubicin (DOX) and magnetic hyperthermia. The toxicological and therapeutic roles of FeONPs is further discussed in Table 2.

### 7.4. Manganese Nanoparticles

Different oxidation states of manganese (Mn) give rise to the formation of MnO_2_, Mn_2_O_3_, and Mn_3_O_4_ forms [178]. Mn_3_O_4_NPs (MnNPs) are recognized as very significant nanomaterials as they possess excellent electrochemical behavior. Proper investigation about the adverse consequences of manganese nanoparticles (MnNPs) on human health has not yet been satisfactorily done as these particles are increasingly used in industrial and biomedical fields. The role MnNPs on ER stress mediated ROS generation, glutathione (GSH) variation, Ca^2+^ imbalance, and apoptosis has been studied on hippocampal neurons. It has been found that these NPs cause the depletion of GSH and increase oxidative stress and calcium level. The expression of ER stress related proteins like caspase (-3, -9, -12), PERK, EIF 2α, GRP78, and GADD153 is enhanced with the introduction of MnNPs [179].

The MnNPs possess a very small size and large surface energy. These NPs possess higher biological activities and can easily enter within the cells by free penetration. This NP entry occurs through the receptor mediated endocytosis, which actively interact with the intracellular components [180]. In addition, the different intracellular MnNP interactions leads to the protein dysfunctions, DNA damage, signaling pathway interference, and excessive ROS production [181,182].

The different growth inhibition assays on various microorganisms including yeast have confirmed the role of MnNPs as inhibitory due to ion dissolution, ROS production, and mitochondrial damage [183]. In addition, these NPs upregulate the UPR genes, thus mediating its toxic effects through the ER stress. As MnNPs impair ER function, it blocks invertase secretion, which is followed by diminished sucrose absorption and results in slowed down cell growth. Furthermore, these NPs lead to decreased protein secretion required for the overall growth [184].

### 7.5. Titanium Oxide Nanoparticles

In recent years, different studies on titanium oxide (TiO_2_) nanoparticles (TiO_2_NPs) have highlighted its role in biomedical applications like biosensors, drug delivery system, cancer therapy, cell imaging, and genetic engineering and related biological experiments [185]. All these observations also support its toxic role both in in vivo and in vitro systems. These particles (<100 nm diameter) generate free radicals and enhance DNA adduct formation in lung fibroblasts [186]. TiO_2_NPs induce mitochondrial injury via ROS production in A549 cells and have been reported to produce inflammation and genotoxicity in animal models and different cell lines [187,188].

TiO_2_NPs induce ER stress in human bronchial epithelial cells by promoting IRE-1α phosphorylation, elevating the levels of CHOP and GRP78/Bip expression, followed by Ca^2+^ homeostasis disruption. As the Ca^2+^ level is regulated by mitochondria cooperatively with ER, it gets translocated from ER to mitochondria through mitochondria associated-ER membranes [189].

TiO_2_ photocatalyst NPs have been widely used in cancer therapy based on their biocompatible and photocatalytic properties [187]. The light-driven TiO_2_NPs interfere with the cellular functions through ROS production, so exerting toxicity in cancer cells [190]. These NPs can trigger the malignant cells through ROS mediated apoptosis, which also affects the adjacent cells within these tissues [191,192]. The ROS-mediated accumulation of unfolded/misfolded proteins in ER triggers the adaptive intracellular stress response to minimize this stress by either correcting or degrading the misfolded proteins within the ER lumen [193]. These NPs trigger apoptosis through the induction of ER sensor polypeptides, which include PERK, ATF6, eIF1, and XBP1, and their downstream signaling pathways like the CHOP pathway [194]. The induction of these pathways is also reported in different diseases such as glioma cells, liver cancer, and breast cancer [195,196]. The physicochemical properties of TiO_2_NPs and their specific role for inducing ER stress is further summarized in Table 3.

### 7.6. Zinc Oxide Nanoparticles

Zinc oxide (ZnO) nanoparticles (ZnONPs) have a great potential for various applications of consumer products, so it is therefore crucial to assess their possible health risks. These NPs are extensively used in sunscreens, cosmetics products, and superior textiles. Aside from the direct use of these NPs, they are used as self-charging and in different electronic devices. Therefore, there is a great possibility of human exposure and the health impacts at each stage of their production and use. This subject of ZnONP use remains a concern for different health issues. In addition to the pulmonary damage, ZnONP exposure is also effectively correlated with the increased incidences of cardiovascular diseases and some allergic reactions [197].

ZnONPs induce their toxicity to the cells through the release of toxic Zn^2+^, which induces the production of ROS [198]. Human umbilical vein endothelial cells (HUVECs) have been used to investigate the cellular responses and ER stress provoked by these NPs. It has been found that dissolved Zn^2+^ ions are the most significant factors accountable for their cytotoxicity in HUVECs. ZnONPs, even at a noncytotoxic concentration of 120 μM, induce substantial cellular ER stress response with enhanced expression of spliced CHOP, caspase-12, and XBP1 at the mRNA levels. In addition, the associated ER marker proteins such as CHOP, p-PERK, BiP, GADD34, p-eIF2R, and cleaved Caspase-12 are expressed at the protein levels [155,197]. It has been further observed that 240 μM ZnONPs quickly reduce the ER stress response before it induces apoptosis. Furthermore, it has been demonstrated that ZnONPs trigger the ER stress-responsive pathways, which could be a novel and sensitive end point strategy for nanotoxicological study. The physicochemical properties of ZnONPs and their specific role for inducing ER stress is further summarized in Table 3.

### 7.7. Quartz and Silica Nanoparticles

Our entire atmosphere is full of nanominerals, so it is important to study the role of these nanominerals including quartz nanoparticles (QNPs), on our health [199]. Quartz is a member of the silicate family of minerals, which is crystalline and consists of 70–80% silica arranged in tetrahedral SiO_4_ units. The inhalation effects of quartz particles are well documented to lead to the development of different medical conditions including silicosis and lung cancer [200,201]. Numerous cytotoxicity studies of silica nanoparticle (SiNP) exposure in work places support the possibility of health issues through oxidative stress and inflammation. Different studies on animal models have already shown the damaging effects of SiNPs on proinflammatory stimulation, nucleoplasm, and the formation of fibrotic nodules [202].

Some recent findings have clearly shown that QNPs induce oxidative stress, cytotoxicity, and an inflammatory response that ultimately leads to ER stress and cell death. During stressful conditions of a cell, induced by some foreign factors, the ROS homeostasis gets disturbed, leading to the activation of different stress-related pathways. Excess ROS leads to the production of pro-inflammatory markers like IL-6, IL-1β, TNF-α, and IFN-γ leads to inflammasome activation within the cells after prolonged QNP exposure [203,204]. The role of QNPs as ER stress inducing agents have been confirmed while studying their effects on the cultured lung A549 cell line [205]. In addition, QNPs have been reported to induce intracellular Ca^2+^ level and ER stress responsive marker proteins that result in mitochondrial damage [206,207]. The increased Ca^2+^ level within the cytosol enhances the mitochondrial membrane permealization and ultimately triggers apoptosis [208,209]. Aside from this, the QNP induced ER stress leads to dimerization of PERK proteins after getting released from GRP78 and gets autophosphorylated and induces the phosphorylation of eIF2α. QNP induced ER stress also leads to ATF4 translocation within the nucleus that activates the Caspases and CHOP proteins and induces apoptosis. In addition to ER stress, QNPs also lead to apoptosis, mediated by Caspases and JNK, followed by impairment of mitochondria [210,211].

Silica (SiO_2_) NPs are used to arouse oxidative stress within the cells, thus resulting in cytotoxicity, which is a time and size dependent phenomenon [212]. These NPs also lead to an imbalance of Ca^2+^ homeostasis and membrane damage [213,214]. It has been reported that SiO_2_-NPs also induce the pro-inflammatory response and genotoxicity in CaCo-2 cell lines [215]. These NPs have also been used to induce inflammation and oxidative stress within human macrophages and lung epithelial cells [216,217].

Recently, it has been reported in human hepatoma cells that SiO_2_-NPs promote oxidative stress through the ER stress [218], and leads to the induction of TNF-α and activates MAPK pathways [219]. SiO_2_-NPs induced ER stress also leads to the activation of NF-κB, leading to the expression of interferons [220]. It has been further reported that SiO_2_-NPs persuade the expression of MAP kinase regulated transcription factors like CYMC and CJUN [219]. Furthermore, it has been found that SiO_2_-NP exposure affects different pathways like the expression of the ATF-4 target gene GADD34, and the expression of genes belonging to the MAPK signaling pathways. In addition, these NPs also affect the expression of DNAJB8, IL-8 as well as the pro-apoptotic gene and CHOP expression.

### 7.8. Graphene Oxide Nanoparticles

Graphene oxide nanoparticles (GONPs) are biocompatible and biodegradable, which have been targeted to ER by conjugating with the dansyl moiety by using the ethylene diamine linker. The ER surface contains sulfonamide receptors that have a binding affinity with the dansyl moiety. In addition, the dansyl moiety possesses a fluorescent nature, which helps to track its location within the ER in cancer cells [221,222]. Recently, GONPs have been loaded with doxorubicin (DOX) and cisplatin individually with almost 79% drug loading efficiency.

The drug DOX inhibits the IRE-α of the UPR [223], and cisplatin binds with different proteins like PDI and calreticulin, which reside in the ER and induce the stress [224,225]. Once the GONPs loaded with DOX and cisplatin are internalized within the ER, it leads to the CHOP expression many fold. In addition, these NPs induce the expression of GRP78, which indicates the onset of ER stress. GONPs also induce the formation of autophagosomes and autophagy, which is validated by the expression of LC3B as a marker of autophagy [226,227]. Furthermore, these GONPs alone and in combination with chloroquine exhibited remarkable efficacy in cell killing in lung, breast, and triple negative breast cancer cells. The results of Panday et al. (2020) clearly demonstrate that graphene oxide ER-specific NPs can be used as effective tools to rouse UPR signaling, and so can lead to future cancer therapeutics [228].

### 7.9. Lipid Nanoparticles

The different types of polymeric NPs including the lipidic NPs have been recently used to target the ER for different purposes including the triggering of ER stress in cancer cells [221]. Different cancer cells have been exposed to small molecules and peptide-based self-assembled nanomaterials to provoke ER stress [229,230]. Recently, a NP-conjugate has been engineered for the ER localization consisting of a dodecyl amine, tosyl group (for ER localization), and naphthalimide moiety (a fluorescent label for subcellular localization) comprising tanespimycin or 17-AAG (HSP 90 inhibitor), which is used to induce ER stress. These NPs with a 158 nm diameter are endocytosed by caveolin-mediated intake by HeLa cells and are transported to the ER. These 17AAG-ER-NPs promote the ER stress in addition to nuclear DNA damage, resulting in cell cycle arrest during G2/M phase. These consequences are followed with enhanced apoptosis compared with the free 17-AAG. Furthermore, 17AAG-ER-NPs have been suggested as an efficient platform tool to understand oncogenesis in detail as well as in future-generation cancer therapy [222].

### 7.10. Enzyme Assembly Based Nanoparticles

As ER targeting is considered as a promising future strategy for cancer management, the specific disruption of ER within cancer cells is still a great challenge to be worked out [231]. The current targeting of small molecules like thapsigargin and tunicamycin to ER lacks the proper cell selectivity. These drugs possess strong neurotoxicity, but the lack of specificity hinders their clinical applications. One way has been resolved to target these drugs to the ER by conjugation with toxins like the Shiga toxin [232]. However, this conjugate still faces a remarkable problem, which is to manage the endosomal/lysosomal escape. Therefore, it is very important to manage the novel ER targeting strategy, lysosomal escaping potential, and proper cancer cell specificity. To meet all the criteria of specific ER targeting, the use of enzyme-instructed self-assembly (EISA) has been explored to achieve a proper spatiotemporal control [233].

EISA is an active process commonly used to regulate the proteins and small molecules. Application of EISA for lipids, sterols, peptides, or carbohydrates has exhibited great promise for inhibiting cancer cells, so plays a potential role in cancer therapy [234,235]. As only specific enzymes are enriched in tumor cells that are confined at specific locations, EISA can localize the supramolecular assemblies at the position of these enzymes, and the resulting assemblies efficiently reduce the diffusion and significantly enhance the diffusion-limited interactions [236]. This novel application of EISA is an innovative strategy to target different cell organelles like the nucleus, mitochondria, and cell membrane to boost the efficient accumulation of small molecules to minimize drug resistance [237,238]. This strategy has strongly favored the use of EISA to specifically target the ER in cancer cells for future cancer management.

### 7.11. Carbon Nanotubes

Carbon nanotube (CNT) exposure to HUVECs has been observed to induce oxidative stress as some antioxidants have been reported to alleviate its genotoxicity and cytotoxicity attitude [239,240]. The toxicity of CNTs with shorter diameter and longer length have been reported to be more powerful compared to their other dimensional parameters. These nanotubes have been more toxic to HUVECs as they activate ER stress. Carboxylation of CNTs further increases their cytotoxicity behavior due to the blockade of autophagic flux [241].

CNTs significantly increase oxidative stress due to the decreasing level of GSH, which provokes increased ROS. In addition, CNTs promote ER stress by inducing ER stress biomarkers like CHOP, pCHOP, and DDIT3. Previously, these findings have also been observed as CNTs induce the ER stress in *C. elegans* and cultured human cells [242]. All types of CNTs have been found to considerably downregulate the pro-survival ER stress gene XBP-1s. This phenomena is supposed to decrease the level of anti-apoptotic protein BCl-2 and the enhancement of caspase-3 and caspase-8. This up/down regulation of specific proteins finally leads to the decreased cellular viability. Furthermore, it has been found that pristine hydroxylation and carboxylation of CNTs are equally cytotoxic to HUVECs and all types of CNTs activate the ER stress signaling pathways [243].

## 8. Summary

The vast majority of evidence has now interpreted the role of ER stress response in tumorigenesis and cancer resistance. Some interesting results have clearly shown the innovative possibility of targeting UPR transduction components for cancer therapy by overcoming severe drug resistance. Different researchers have demonstrated the role of diverse metal and non-metal based NPs and other nanocomplexes by triggering ER stress, which mediates the anticancer activity. Up until now, dozens of NPs have been found to possess a novel anticancer property. Even though different NPs possess structural similarities, they activate ER stress through different mechanisms such as redox mediators, Ca^2+^ trafficking, and ROS generation. These different NPs induce ER stress in diverse ways as they can act as proteosome inhibitors, photosensitizers, enzyme inhibitors, and Ca^2+^ trafficking modulators, etc. The most common feature of NPs is the disruption of redox homeostasis as an anticancer activity. The metal complex NPs like CuNPs, PtNPs, and AuNPs enable the metal centers to act as electrophiles. The ER stress in response to such NPs targets the redox regulatory enzymes like PDI and thioredoxin-1 (TRX). Overall, these NPs exhibit outstanding power, even at nanomolar concentration and noteworthy in vivo cancer-reduction abilities. These nanoformulations have also been found to possess more selectivity toward cancer cells. Together, the use of these NPs have demonstrated a great potential as ER-targeting antitumor agents.

The signaling pathways activated by NP-mediated ER stress are not fully understood as it involves molecular mechanisms with dualistic functions in cell survival and death. Thus, understanding how these ER stress pathways signal cell death or prevent it from such steps, comprises a major challenge for future investigations and requires to define a validation for drug design and applications. The challenge of specific cancer treatment in the near future is in the development of drugs targeting the cytoprotective functions of the UPR, and leaving intact or accelerating its pro-apoptotic power. In addition, the actual mechanism that decides the ER specific NPs in eliciting the UPR dependent toggling between the pro-survival and pro-apoptotic signaling cascades needs to be fully comprehended. An in depth focus on this core area of research will be helpful in boosting up developing novel pro-drug candidates to exploit ER stress for triggering pro-death pathways in different cancers. The necessity of a deeper understanding of cancer biology, employment of proper regulatory measures, and advancements in nanoparticle technology will definitely speed up the possible mainstream cancer treatments in the near future.

## Figures and Tables

**Figure 1 molecules-25-05336-f001:**
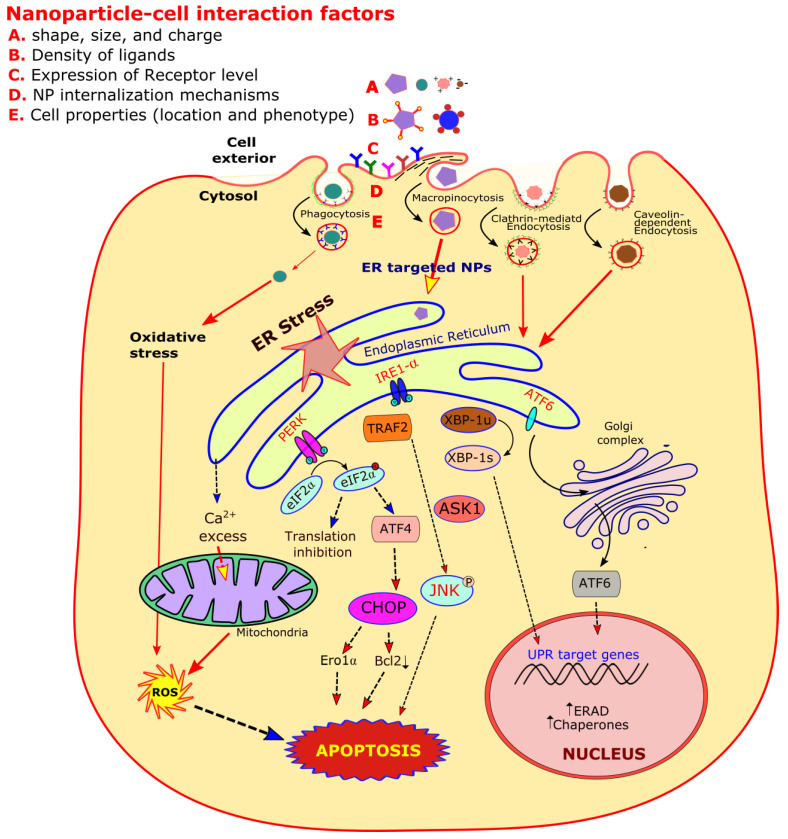
A diagrammatic representation of endoplasmic reticulum (ER) stress and oxidative stress facilitated signaling pathways induced by different types of nanoparticles (NPs). The NP-cell interaction depends on their shape, size, charge, and ligand density. This interaction also depends on cell membrane receptor types, internalization mechanisms, and other cell properties.

**Figure 2 molecules-25-05336-f002:**
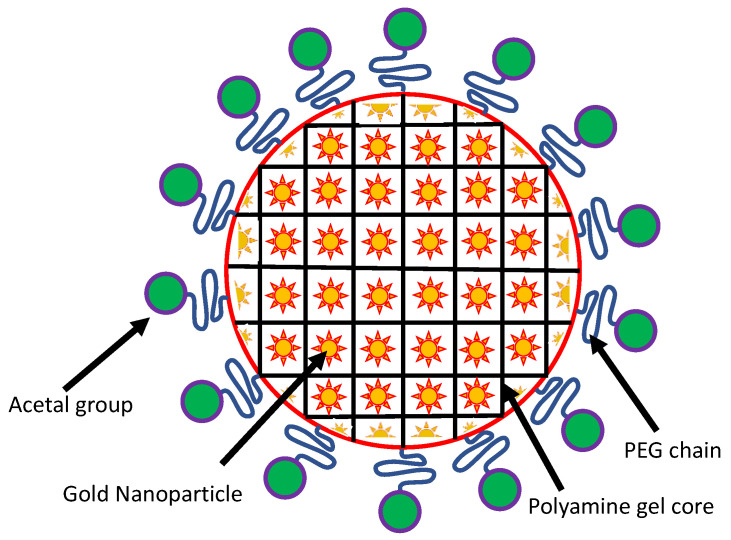
Schematic representation of the AuNP nanogel. The AuNPs are embedded in a core of polyamine gel, which are PEGylated with terminal acetal groups.

**Table 1 molecules-25-05336-t001:** Some common types of diseases induced by endoplasmic reticulum stress.

S. No.	Disease	Role of ER Stress	References
1.	Type 2 diabetes	ER stress is induced by obesity, gluconeogenesis gets affected by ATF6. The pancreatic β-cell death is induced by hyperglycemia and free fatty acids through CHOP signal	[32]
2.	Parkinson’s disease	Substrate of Parkinson accumulation in ER, leads to ER stress	[33]
3.	Alzheimer’s disease	The CHOP cascade gets activated by mutant presenilin	[34]
4.	Atherosclerosis	Smooth muscle and endothelial cell death is mediated by hyperhomocysteinemia, oxidized phospholipids and cholesterol loading which leads to CHOP cascade. Macrophage death is induced by CHOP cascade which is generated by atherosclerosis related stimuli	[35]
5.	Alcoholic liver disease	The induction of GRP78 and CHOP occurs by alcohol consumption	[36]
6.	Non-alcoholic fatty liver disease	SREBP-1c is induced by ER stress	[37]
7.	HBV and HCV infection	GRP78 and GRP 94 are induced by HBV, while IRE1-XBP1 pathway is suppressed by HCV	[38]
8.	Amyotrophic lateral sclerosis	ER stress is activated by mutant SOD1 aggregates	[39]
9.	Cancer	Different cancers lead to the induction of GRP78 and XBP1	[40]
10.	Ovarian cancer	The ovarian cancer patients have increased expression of GRP78	[41]
11.	Liver cancer	In human hepatocellular carcinoma tissues and human hepatocellular carcinoma cells (SMMC7721), the enhanced expression of Grp78 promotes the invasion of hepatocellular carcinoma both in vivo and in vitro	[42]
12.	Prostate cancer	The hormone-resistant prostate cancer cells promote the enhanced expression of Grp78 to the cell surface, which can be further elevated by ER stress in human prostate adenocarcinoma hormone-resistant cells, C4-2B	[43]
13.	Lymphoma	The splicing of XBP1 stimulates tumor growth during hypoxic conditions in such patients	[44]
14.	Breast cancer	In human breast cancer tissue and breast carcinoma cell lines (MDA-MB-231, MCF-7, HCC1500 HS578T), and cells show elevated levels of mRNA and protein Bip/Grp78	[45]
		During hypoxia, increased level of ATF4 occurs in MCF7 cell lines	[46]
		Increased expression of unspliced XBP1 mRNA favoring apoptosis of cancer cells and higher levels of spliced XBP1 mRNA increasing cancer survival in human breast cancer tissue	[47]
		The Hormone-resistant breast cancer cells stimulate Grp78 expression on the cell surface, which is further elevated by ER stress in human breast cancer hormone-resistant cells and MCF-7/BUS-10 cell lines	[43]
15.	Colorectal cancer	Increased expression of ATF4 occurs during severe hypoxia in HT29 cells	[48]
		Increased expression of Grp78 on CRC cell surfaces, which promotes CRC cell migration and invasion In human colon carcinoma (SW480, HT29, DLD1, SW620, and Lovo) cell lines	[49]
16.	Brain and central nervous system tumor	Enhanced expression of Grp78 is observed in human brain tumor specimens and glioma cell lines (U87, A172, U251, LNZ308, LN-229 and LN-443)	[50]
		The depletion of XBP-1 dramatically sensitizes U373 cells to viral oncolysis in U373 glioblastoma cells	[51]
		The Inhibition of IRE1α enhances the oncolytic therapy in glioblastoma patient samples	[51]
17.	Pancreatic cancer	The expression of PERK supports the proliferation of beta-cell insulinoma and promotes angiogenesis in human tumor xenograft mice	[52]

**Table 2 molecules-25-05336-t002:** The toxicological and therapeutic effects of different types of nanoparticles.

Nanoparticles	Toxicological Effects	Therapeutic Effects
Aluminum oxide	These NPs used as 1–10 μM in HBMVECs, decreased cell viability, decreased mitochondrial functions, and increased oxidative stress [90]	These NPs were used to penetrated Candida cells, which disrupted the morphological and physiological activity of these cells [91]
	These NPs 50–80 nm in diameter, were used in mammalian cells EZ4U, caused no significant toxic effect on cell viability [92]	These NPs showed good antibacterial, activity against clinical isolates of *P. aeruginosa* was significant [93]
	The NP were used as 160 nm in size in HMSC, caused decreased cell viability [94]	The NPs were effective against gram-positive and gram-negative bacteria [95]
	The NPs were used to check rat blood cells comet micronuclei, caused dose-dependent genotoxicity [96]	The NPs were used as anti-cancer therapy, cytotoxic agents to induce cell death in human prostate cancer cells, and for boosting the efficacy of cancer vaccines [97]
	These NPs (50 nm) were used as 0–5000 μg/mL to induce comet DNA damage [98]	The NPs were used as leishmania vaccine to induce autophagy in macrophages and as potent vaccination adjuvant [99]
Copper oxide	These NPs were used in human lung epithelial cells, caused decreased cell viability, increased LDH expression and enhanced lipid peroxidation [100]	The NPs were used in MCF-7 breast cancer cell lines for their cytotoxic effect therapeutics [101]
	These NPs were used as 0.002–0.2 μg/mL in lung cancer cells, lead to decreased cell viability [102]	These NPs caused skin cancer cells A-375 apoptotic cell death by the activation of caspase-9 [103]
Gold	The NPs caused enhanced lipid peroxidation, oxidative damage and upregulation of stress response genes, and protein expression [104]	The NPs caused the Inhibition of angiogenesis and acted as promising candidates for the drug delivery systems and in cancer therapeutics [105]
	These NPs induced a significant toxicity, effectively entered the cytoplasm and nucleus, leading to the damage of cellular and nuclear membranes [106]	These NPs were used for the treatment of rheumatic diseases including juvenile arthritis, psoriasis, palindromic rheumatism, and discoid lupus Erythematosus [107]
	As 5-nm size, induced the cytotoxicity at 50 mM, whereas no toxicity was observed when used as 15-nm NPs. This shows the size-dependent toxicity of NPs [108]	These NPs increased the apoptosis in B-chronic lymphocytic leukemia (BCLL) treatment-suffering patients [109]
	The chitosan-functionalized AuNPs induced the cytotoxicity and pro-inflammatory responses. This indicates the charge-dependent toxicity [110]	The NPs were used as surface coating for different biomedicine applications such as dressing fabrics, implants, glass surfaces [111]
Silica	These NPs used as 10–100 μg/mL in human bronchoalveolar carcinoma cells showed enhanced ROS production, increased LDH expression and higher malondialdehyde formation [112]	As silica-gold nanoshells and gold nanorods were used for tissue stimulating phantoms during photothermal therapy [113]
	These NPs were used in hepatocellular carcinoma cells (HepG2), showed enhanced ROS production and mitochondrial damage due to increased oxidative stress [114]	The Silica-gold nanoshells were used in human breast carcinoma cells (in vitro) and transmissible venereal tumor (in vivo) as a photothermal therapy agents [115]
Silver	These NPs were used in BRL 3A cell lines and resulted in decreased cell viability, increased LDH expression due to enhanced ROS production [116]	These NPs capped with polyvinyl-pyrrolidone encapsulated in polymer Nanoparticles Human Glioblastoma. Astrocytoma epithelial-like Cell line (U87MG) (in vitro); Swiss mice and severe combined immunodeficiency mice bearing U87MG tumors (in vivo) Therapeutic Evaluation [117]
	These NPs were used as 0–20 μg/mL in human alveolar cell lines, resulted in decreased cell viability due to increased ROS production [118]	The silver nano-shell with a carbon core were used in prostate adenocarcinoma cell line model as photothermal ablation or radiation enhanced therapy [119]
	These NPs (20–40 nm) in size were used in human leukemia cell line WST-1 and resulted in decreased cell viability and the increased expression of LDH [120]	These NPs acted as the excellent candidates for bioimaging and act as good anticancer agents [121]
Zinc oxide	These NPs were used in human colon carcinoma cells, resulted in increased oxidative stress, decreased cell viability and the expression of more inflammatory biomarkers [122]	These NPs when used in murine cell lines showed cytotoxic effects [123]
	Larger NPs (307–419 nm) in size were used in in human cervix carcinoma cell line (HEp-2), enhanced the DNA damage and decreased the cell viability [124]	These NPs were regarded as a possible treatment for cancer and autoimmune diseases and were found to be involved in specific killing of cancer cells and lead to the activation of human T cells [125]
	These NPs resulted in decreased cell viability due to increased DNA damage and increased ROS production and leading to apoptosis [126]	These NPs were used in bioimaging, drug delivery, gene delivery, and as zinc-based biosensors [127]
	These NPs were used in human hepatocytes HEK 293 cell line, reduced cell viability, increased mitochondrial damage due to higher oxidative stress [128]	These NPs were used to prevents herpes, by stopping the viral entry and infection [129]
	These NPs (<20 nm) were used as 100 μg/mL in human bronchial epithelial cells showed decreased cell viability, LDH release due to enhanced oxidative stress [130]	These NPs were used to prevents the helminth infection as it disrupts the electron transport system and inhibiting the ATP production, so stopping the contractile movement of the parasite [131]
Iron oxide	These NPs were used in murine macrophage cells and resulted in decreased cell viability [132]	These NPs as superparamagnetic NPs were coated with silica-gold nanoshells and used in head and neck cancer cell lines and resulted in overexpression of EGFR and were used for photothermal therapy [133]
	The NPs as (100–150 nm) in sized used as 0.1 mg/mL in human macrophages resulted in decreased cell viability [134]	These NPs were used in prostate cancer, were magnetic field responsive for thermal ablation [135]
	These NPs were used in human hepatocellular carcinoma cells resulted in decreased cell viability [136]	These NPs as aminosilane-coated, were used for thermotherapy during brain tumors [137]
	The NPs (20 nm) were used as 0.1 mg/mL in rat mesenchymal stem cells resulted in decreased cell viability [138]	These NPs as starch-coated were magnetically guided for mitoxantrone tumor angiogenesis [139]
Titanium oxide	These NPs were used in mouse models, resulted in enhanced DNA damage and resulted in genotoxicity [140]	These NPs were used in CT26 and LL2 mouse cancer to increase oxidative stress [141]
	These NPs were used as 10–50 μg/mL in human lung cells resulted in enhanced oxidative stress, more DNA adduct formation and increased cytotoxicity [142]	These NPs were used as efficient drug delivery systems and in photodynamic therapy of tumors [143]

**Table 3 molecules-25-05336-t003:** Different types of metal and non-metal NPs possessing diverse physico-chemical properties that have specific roles in ER stress provocation and special applications, which were investigated to study their role in the management of cancer and other diseases.

Nanoparticle	Physicochemical Properties and Related Studies	Specific Role in ER Stress Induction	Application	Reference
AgNPs	Typical size of 120 nm, negatively charged ZFL cells (in vitro); exposure on zebrafish (in vivo) 0.05–0.5 mg/mL for 6–24 h (in vitro); 0.1–5 mg/mL for 24 h (in vivo)	Increase in GRP78, ATF6, and XBP-1s protein expression or mRNA synthesis.	ROS induction, ER stress response, apoptotic and inflammatory pathways activation.	[72]
	Average size 20 nm, negatively charged 16HBE cells (in vitro); mice (in vivo) 2 μg/cm^2^ exposure from 4 to 24 h (in vitro); 0.1–0.5 μg/g (in vivo)	Increased p-PERK, XBP-1s, p-IRE1α, CHOP, GRP78, p-eIF2α protein expression or mRNA synthesis	Cellular response on different cell lines, to know the mechanisms of action in various cellular systems, cellular activation of different signaling molecules	[244]
	10, 50 and 100 nm used in HepG2 cells as 1 μg/mL dose for 24 h	Increase in CHOP protein	Can be used as cytotoxic agents on mice liver primary cells and also in human liver HepG2 cells	[245]
	15 nm in size, negatively charged in THP-1 cells as 1–25 μg/mL for 1 or 24 h	Increase in p-PERK protein and ATF6 degradation	Redox active particles can induce toxicity mediated through ROS production and increases oxidative stress	[246]
	≤100 nm size, used in Human Chang liver cells as 4 μg/mL for 3–24 h	Increase in ER tracker staining and protein levels of p-IRE1, p-PERK, ATF6, peIF2α, XBP-1s, GRP78, and CHOP	Increased concentrations of these NPs induce substantial cytotoxicity, DNA damage and apoptosis.	[247]
	Size 2 nm to 10 nm; negatively charged used in MCF-7 and T-47D cells	Increase in p-eIF2α, p-PERK, CHOP, p-IRE1α, and ATF4 proteins	Possess anti-cancer activity, DOX + AgNPs can induce conformational changes on DNA	[248]
AuNPs	Size about 12 nm and citrate-capped are negatively charged used in HUVECs as 8 μg/mL from 2 to 35 d	Enhanced XBP-1s mRNA production	Accumulates at steady exposure of lower (non-lethal) dose and causes no measurable cell death while leading to elevated ER stress.	[249]
	Size 20 to 70 nm and negatively charged used in human neutrophils as 100 μg/mL for 3 h	Increased p-IRE, p-PERK, and ATF6 proteins synthesis	PEG-AuNPs can be efficient drug delivery vehicles, and exhibit least adsorption of proteins and slight size and charge deviation when used in whole blood	[250]
	20 nm in size, citrate coated AsPc1 cells	Enhanced IRE-1α and CHOP proteins synthesis	Sensitization of pancreatic cancer cells by the pre-treatment with these NPs in addition to gemcitabine in colony forming and viability assays	[251]
	Size of 1–6 nm and 15–20 nm used in K562 cells	Increased the ER stress related proteins checked by proteomic assay	These NPs can be used to diminish the growth and provoke strong apoptosis in human chronic myeloid leukemia cells	[252]
PEGylated nanogel with AuNPs	Used in SCCVII and A549 cells	Increased GRP78, IRE-1α, p-PERK protein synthesis	Favors the radiosensitization of cells to increase the apoptosis and ER stress provoked DNA repair capacity	[253]
ZnO NPs	Size about 100 nm used in HUVEC at a dose of 240 μM for 4–24 h	Augmented CHOP, p-PERK, XBP-1s, p-eIF2α, HSP proteins or mRNA production	Activates the ER stress-responsive pathways	[197]
	Size between 30 nm to 90 nm, bulk 100–200 nm used in mice at 100 mg/kg/d for 3 d	Increase in eIF2a, PERK, ATF4, JNK, CHOP, GRP94 mRNA in livers	It disrupts seminiferous epithelium of the testis and decreases the sperm density in the epididymis	[254]
	The size is about 70 nm; positively charged used in MRC5 cells at 25 and 50 μg/mL for 16 h	Increases CHOP and ERN1 mRNA synthesis	Oxidative stress is promoted, which causes cytotoxicity and genotoxicity in human lung fibroblasts in vitro and in *D. melanogaster in vivo*	[255]
	The size is <100 nm and negatively charged used in mice as gavage for 90 d (200, 400 mg/kg)	It causes the swelling of ER; increases GRP 78/94, XBP-1, and PDI-3 mRNA synthesis, CHOP and p-JNK protein production in liver	The relationship of the dosage and organs affected as pancreas, stomach, eye, and prostate gland	[256]
ZnS NPs	Size between 50 and 100 nm used in mice retinal pigment epithelial cells	Inhibited GRP78 and CHOP protein synthesis	Can be used as anti-age-related macular degeneration	[257]
Fe_3_O_4_ NPs	Hydrodynamic size about 26 nm, negatively charged and used in RAW 264.7 cells as 6.25–50 μg/mL for 24 h	Increase in CHOP mRNA, CHOP, p-IRE1α, IRE1α protein synthesis	Pyroptosis demonstration and IL-1β synthesis, safety evaluation of metal oxides	[258]
	15–20 nm in size, PLGA coated about 300 nm in size used in MCF-7 cells as 100 μg/mL for 24 h.	Disrupts and disperse ER	Gemcitabine loaded NPs demonstrate as multifunctional drag cargo system, can be used during radiosensitization investigations	[259]
TiO_2_ NPs	Size as P25 (24 nm), and scrolled nanosheets (L/W 178/9), nanoneedles (L/W 45/15), isotropic NPs (29 nm) used in HUVECs cells as 2 μg/cm^2^ for 1–24 h	Increase in ERdj4, CHOP, HERPUD1 mRNA (scrolled nanosheets	Such NPs can be used to increase the ROS production having a central role in the induction of receptor expression	[260]
	Size as 19.3 ± 5.4 nm, used in mice, inhaled to 2.5, 5.0 and 10.0 mg/m^3^ NPs for a span of 28 d	Causes the swelling of ER and increases CHOP, GRP78, and p-IRE1α protein synthesis in lungs	Can be used as toxicological index that acts as a benchmark for assessing the risks to human health	[261]
	Hydrodynamic size of about 250 nm, the anatase: rutile ratio of 8:2 used in 16HBE14o-lung cells as 50 and 100 μg/mL for 24 and 48 h	Increases the CHOP, GRP78, IRE-1α, and p-IRE-1α protein synthesis	The Anatase TiO_2_ NPs induces increased inflammatory responses as compared with other TiO_2_ particles	[262]
Cadmium telluride (CdTe) quantum dots (QDs) (CdTeQDs)	About 4 nm in size and negative charged used in HUVECs cells as 10 μg/mL for 24 h.	Lead to the dilation of ER and protein synthesis increase of GRP78/95, CHOP, ATF4, p-PERK, peIF2α, and p-JNK.	The toxicity of QDs can act as potential cardiovascular risk factors	[263]
Poly [lactic-co-glycolic acid] (PLGA) NPs containing γ-oryzanol	Size about 214.8 nm with negative charge used in obese ob/ob mice	It reduced the CHOP, ERdj4, and XBP-1s mRNA synthesis	Can be used during metabolic diseases treatment	[154]
Poly [lactic-co-glycolic acid] (PLGA) NPs containing LY294002	NPs with an average size of 98.9 ± 2.64 nm and used in H157, H460, H1650, and NL20 cells	It leads to the accumulation in ER; increased GRP78, CHOP, and p-JNK proteins	These NPs act as surfactant-free formulation of PLGA and possesses a promising anticancer activity	[264]
CeO_2_ NPs	Average size of 7 nm used in MCP-1 transgenic mice	It suppresses the GRP78, PDI, and HSP mRNA synthesis	These NPs slow down the advancement of cardiac dysfunction myocardial oxidative stress	[265]
	Used in H9C2 cells	It reduces the PDI and GRP78 proteins synthesis	These NPs are pH responsive with anti-tumoral activities for osteosarcoma	[266]
PEGylated-Phosphatidyl ethanolamine (PE) micelles	Used in MRC-5, A549, 293T cells for ER dilation	Leads to increased IRE-1α, eIF2α, PERK, ATF4/6, XBP-1s and CHOP proteins synthesis in cancer cells	It enhances the sensitivity of most cancer cells to some chemotherapeutic agents	[267]
Gadolinium metallofullerenol [Gd@C_82_(OH)_22_]_n_ NPs	Used in MCF-7 and ECV304 cells	It lead to slowed protein processing in ER and also increased the CHOP mRNA synthesis as reported by DNA microarray	These NPs possess high anti-tumor activity but have low toxicity	[268]
Realgar QDs	These NPs have an average size of 5.48 nm and used in JEC cells	It induces the dilation of ER and increased CHOP and GRP78 mRNA and proteins synthesis	Can be used effectively against human endometrial cancer cells as it leads to ER stress mediated necrosis and apoptotic cell death	[269]
Anodic Alumina Nanotubes (AANTs) loaded with Thapsigargin (TG)	The length is 736 nm ± 460 nm, inner diameter and outer diameter as 33.0 ± 8.0 and 90.0 ± 10.0 nm used in THP-1, HFF, and MDA-MB 231-TXSA cells	It led to increase in IRE1α and GRP78 proteins synthesis and ER tracker staining	It can act as novel biomaterials for clinical cancer therapy as it can act as ER and autophagic delivery systems	[270]
Anodic Alumina Nanotubes (AANTs)	It has the aspect ratio of 7.8 (short), 27.7 (medium) and 63.3 (long) used in MDA-MB-231-TXSA and RAW264.7 cells as 100 μg/mL AANTs for 3d.	It led to increased CHOP protein synthesis and ER-tracker staining and decreased IRE1α protein synthesis as reported by long AANT only.	For the advanced drug delivery applications, it has a promising opportunity as it can control the nanotoxicity of high aspect ratio nanomaterials	[271]
Extremely small size iron oxide NPs (ESION) and MnONPs	About 3 nm (ESION), and 15 nm (MnONPs) used in mice as 2, 5, 10 μg/g for 1 d	Enhanced the expression of CHOP, HSP, GRP78, XBP-1s mRNA or protein in various organs	NPs exposure causes bodyweight loss, increased NO and MDA levels, inflammatory and hyperplastic changes in the lung homogenates	[272]
NH_2_-labelled Polystyrene (PS) NPs	60 nm in size, positively charged used in RAW 264.7, BEAS-2B cells as 5–40 μg/mL up to 16 h.	It leads to misfolded protein aggregates; increases ER-tracker staining and IRE1α protein synthesis	These NPs can play an efficient role in autophagy, safe and novel material design and inhibition of the toxicity	[273]
Chitosan NPs	Average size of 100 nm used in mouse morula-stage embryos as 100 μg/mL for 24–28 h	It leads to increase in GRP78, CHOP, ATF4, PERK, IRE-1α, protein or mRNA synthesis	These NPs lead to blastocyst complications with no or small cavity	[274]
Silica (SiO_2_) NPs	About 250 nm in size and negatively charged used in Huh7 cells as 0.05–0.5 mg/mL for 4 and 24 h	These NPs increased the GRP78 and XBP-1s mRNA synthesis	These NPs lead to ER stress mediated MAPK pathway, and inflammatory reactions initiation in human hepatoma cells	[197]
Polyethyleneimine (PEI) NPs	Used in Neuro2A cells as 3–25 μg/mL for 24 h	It causes increased GRP78, ATF4 and CHOP mRNA synthesis	These NPs cause Neuro2A cells induced cell toxicity in a concentration-dependent manner	[81]
Curcumin NPs	Average size of 50 nm and negative charged used in H9C2 cells	These NPs cause suppression of GRP78 and CHOP proteins	These NPs can prevent myocardial injury	[275]

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
