# Peer review of "Endoplasmic Reticulum Stress Provocation by Different Nanoparticles: An Innovative Approach to Manage the Cancer and Other Common Diseases"

_molecules, 2020, doi:10.3390/molecules25225336_

Round 1
Reviewer 1 Report
Dear Authors,the review is well organized and the topics are covered in an exhaustive mode. There are characters of different types in the text, so it would be a good idea to check the formatting.
Author Response
We are very thankful to you as a reviewer. In our opinion, fulfilling all these valuable comments has definitely improved the quality of this review article.
Comment: Dear Authors, the review is well organized and the topics are covered in an exhaustive mode. There are characters of different types in the text, so it would be a good idea to check the formatting.
Response: Thanks a lot for your appreciation. We have unified the formatting of the text accordingly
Reviewer 2 Report
The review by Khan et al. provides a perspective on the effects of nanoparticles over endoplasmic reticulum, as an innovative diseases treatment.
The review is well written, with 199 references, although I would like to see more recent ones.
The authors must review the journal’s rules for cites and references, as the style is not always the same (for example there are many references with year after authors, while others have year after journal name – 8, 30, 32 etc)
In the Introduction, authors should mention the review procedure that they have followed.
I think that a short discussion about nanoparticle’s size and shape influence on the biological effects would be welcome, beside the size values in table 2 (where some of them are at the upper limit of nanoscale).
Fe3O4 and other iron oxide NPs should also have a place in chapter 6.
Author Response
We are very thankful to you as a reviewers. In our opinion, fulfilling all these valuable comments has definitely improved the quality of this review article.
Comment: The review is well written, with 199 references, although I would like to see more recent ones.
Response: Thanks a lot for your appreciation. We have updated most of the references with latest ones, wherever possible (marked as red color font in the reference section).
Comment: The authors must review the journal’s rules for cites and references, as the style is not always the same (for example there are many references with year after authors, while others have year after journal name – 8, 30, 32 etc)
Response: All the references have now more or less a unified format as: author names, year of publication, journal name, volume, page number.
Comment: In the Introduction, authors should mention the review procedure that they have followed.
Response: At the end of the introduction section, we have now properly mentioned the different sections followed in the review article.
Comment: I think that a short discussion about nanoparticle’s size and shape influence on the biological effects would be welcome, beside the size values in table 2 (where some of them are at the upper limit of nanoscale).
Response: We have properly added a new section about physico-chemical characteristics of nanoparticles and its role on biological systems (page 6). (Marked as a red color font). In addition, Fig. 1 has been modified accordingly on the basis of different types of NPs and its interaction with a cell. The new references have been added which are marked as a green color font in reference section.
Comment: Fe3O4 and other iron oxide NPs should also have a place in chapter 6.
Response: One more section, as Iron oxide nanoparticles and other types of iron oxide NPs have been added (page 13) marked as a red color font in main text. The new references are marked as green color font in reference section.
Reviewer 3 Report
Amjad Ali Khan and coworkers have submitted a manuscript entitled “Endoplasmic Reticulum Stress Provocation by 3 Different Nanoparticles: An Innovative Approach to 4 Manage the Cancer and Other Common Diseases” which is an interesting review article. Authors have reported several important findings on the role of endoplasmic reticulum stress in association with different type of nanoparticles. The presentation of the information through figures and tables are impressive and easy to understand the subject for both scientific and non-scientific readers. I have few comments for the improvement of manuscript-
- English is good but still some large sentences could be rephrased for better understanding. Specifically, there is the tendency of abusing of colloquial that in my opinion are not grammatically correct. I would invite the authors to generally revise the language by a professional English editing service to improve the comprehension of the contents.
- Abstract should be more precise and focused. The central message of this review is missing in the abstract.
- Several new studies on the role of nanoparticles are missing in the text. Authors have cited references from 2007-2008, it should be improved by citing new references. Specially, the importance of ER-specific nanoparticles in cancer should be discussed in brief.
- According to the figure 1, how this mechanism works for all the nanoparticles? Nanoparticles may vary from their size, shape, characteristics, properties and also most of NP are cell specific. This should be more specific for example, silver or gold nanoparticle. What is the role here of oxidative stress or ROS? This should be discussed in figure and text as well. Expand the figure legend by discussing more in details.
- Author have reported the therapeutic role of nanoparticles against several diseases. I did not see reports on the toxic effects of these nanoparticles. My suggestion is to include a table describing both adverse and therapeutic effects of nanoparticles by introducing suitable nanocarriers to target cancer cells with relevant citations.
- Include future perspectives in summary discussing more on the ER-specific nanoparticles for the therapeutics of cancer.
Author Response
We are very thankful to you as a reviewer. In our opinion, fulfilling all these valuable comments has definitely improved the quality of this review article.
Comment 1: English is good but still some large sentences could be rephrased for better understanding. Specifically, there is the tendency of abusing of colloquial that in my opinion are not grammatically correct. I would invite the authors to generally revise the language by a professional English editing service to improve the comprehension of the contents.
Response: Thanks a lot for your comments. We have thoroughly checked the manuscript for any grammatical mistakes and some long sentences have been divided or rearranged accordingly. As per your valuable suggestion, this review article has been proof read quite thoroughly for english editing, as we seek the help from professional editing service to edit this manuscript. All the possible changes in the manuscript about any grammatical mistakes has been marked as a red color font.
Comment 2: Abstract should be more precise and focused. The central message of this review is missing in the abstract.
Response: The abstract has been thoroughly revised and the central message of this review article has been properly addressed. All the possible changes in the abstract has been marked as a red color font.
Comment 3: Several new studies on the role of nanoparticles are missing in the text. Authors have cited references from 2007-2008, it should be improved by citing new references. Specially, the importance of ER-specific nanoparticles in cancer should be discussed in brief.
Response: Some new studies about the role of nanoparticles have been added in the text (page 8) and table 2 (toxicological and therapeutic effects of nanoparticles). All the new references are marked as green color font. Most of the references (2007-2008) have been updated with more recent references, wherever possible (marked as a red color font in the reference section).
Comment 4: According to the figure 1, how this mechanism works for all the nanoparticles? Nanoparticles may vary from their size, shape, characteristics, properties and also most of NP are cell specific. This should be more specific for example, silver or gold nanoparticle. What is the role here of oxidative stress or ROS? This should be discussed in figure and text as well. Expand the figure legend by discussing more in details.
Response: The figure 1 has been properly updated with more information about the different physico-chemical properties of nanoparticles and its interaction with cells through cell membrane. In addition, the figure 1 and the text has been updated about the role of oxidative stress or ROS. In the text also, one new section has been added about the physico-chemical characteristics of nanoparticles and its interaction with biological system (page 6). The figure legend has been properly updated. All the changes are marked as a red color font in the main text. The new references are marked as green color font in reference section.
Comment 5: Author have reported the therapeutic role of nanoparticles against several diseases. I did not see reports on the toxic effects of these nanoparticles. My suggestion is to include a table describing both adverse and therapeutic effects of nanoparticles by introducing suitable nanocarriers to target cancer cells with relevant citations.
Response: The reports about the toxic effects of nanoparticles has been added accordingly in the text. A new table (table 2) has been added accordingly which describes the adverse and therapeutic effects of nanoparticles targeted to cancer cells with relevant citations. All the new information is marked as a red color font in the main text. The new citations are marked as green color font in reference section.
Comment 6: Include future perspectives in summary discussing more on the ER-specific nanoparticles for the therapeutics of cancer.
Response: The summary has been updated accordingly describing the future prospects of ER-specific nanoparticles for the therapeutics of cancer (marked as a red color font).
Round 2
Reviewer 3 Report
Authors have revised manuscript quite well by fulfilling all my comments. Added all the additional materials in the text and described well.